# New 3-Aminopropylsilatrane Derivatives: Synthesis, Structure, Properties, and Biological Activity

**DOI:** 10.3390/ijms24129965

**Published:** 2023-06-09

**Authors:** Sergey N. Adamovich, Igor A. Ushakov, Elizaveta N. Oborina, Svetlana V. Lukyanova, Vladislav Y. Komarov

**Affiliations:** 1A.E. Favorsky Irkutsk Institute of Chemistry, Siberian Branch of the Russian Academy of Sciences, 1 Favorsky Street, 664033 Irkutsk, Russia; 2Irkutsk Antiplague Research Institute of Siberia and Far East, 78 Trilisser Street, 664047 Irkutsk, Russia; 3A.V. Nikolaev Institute of Inorganic Chemistry, Siberian Branch of the Russian Academy of Sciences, 3 Lavrentiev Prospekt, 630090 Novosibirsk, Russia

**Keywords:** 3-aminopropylsilatrane, acrylates, aza-Michael reaction, mono- and diadducts, biological activity

## Abstract

The biologically active compound 3-aminopropylsilatrane (a compound with a pentacoordinated silicon atom) underwent an aza-Michael reaction with various acrylates and other Michael acceptors. Depending on the molar ratio, the reaction yielded Michael mono- or diadducts (11 examples) containing functional groups (silatranyl, carbonyl, nitrile, amino, etc.). These compounds were characterized via IR and NMR spectroscopy, mass spectrometry, X-ray diffraction, and elemental analysis. Calculations (using in silico, PASS, and SwissADMET online software) revealed that the functionalized (hybrid) silatranes were bioavailable, druglike compounds that exhibited pronounced antineoplastic and macrophage-colony-stimulating activity. The in vitro effect of silatranes on the growth of pathogenic bacteria (*Listeria*, *Staphylococcus*, and *Yersinia*) was studied. It was found that the synthesized compounds exerted inhibitory and stimulating effects in high and low concentrations, respectively.

## 1. Introduction

It is common knowledge that both the introduction of a silicon atom into an organic molecule and the isosteric substitutions of carbon for silicon can change and improve the physical and chemical properties of a compound, as well as their biological, physiological, and pharmacological activities [1,2,3]. For instance, sila-isosteres exhibit better stability, lipophilicity, bioavailability, efficiency, and safety compared with their analogs. Generally, the strategy of bio-sila-isosterism has promising outcomes, and Si-containing structures can become highly effective agents against severe diseases, such as hypertension, diabetes mellitus, HIV, and cancer [4,5,6,7,8,9,10,11,12,13,14,15,16].

Over the last few decades, bioorganosilicon chemistry has witnessed a number of breakthrough results [17,18]. This is also true of intra-complex compounds of pentacoordinated silicon and silatranes R-Si(OCH_2_CH_2_)_3_N. It should be emphasized that, depending on the R substituent and as a result of the presence of an intramolecular coordinate bond (Si←N) and a unique tricyclic structure, silatranes exhibit high and diverse physiological and pharmacological activities, including antifungal, antimicrobial, antiparasitic, and anticancer properties [19,20,21,22,23,24,25,26].

To enhance the pharmacological activity of drugs, molecular hybridization (MH) is employed. According to this method, the synthesis of a chemical compound involves the combination of two pharmacophoric units into one molecule [27,28,29,30]. It is known that the MG strategy can be applied to the combination of silatranes with other pharmacophores [31,32,33,34,35,36,37].

In addition, silatranes possess special physical–chemical properties. For example, as a result of their slow and controlled hydrolysis, they can, ideally, modify (functionalize) glass, polymer, metal, and other surfaces, in contrast to the associated alkoxysilanes (Figure 1) [38,39,40].

Thus, silatranes are very promising compounds for agriculture, medicine, and industry. The above properties are true for commercially available 3-aminopropylsilatrane, NH_2_-(CH_2_)_3_-Si(OCH_2_CH_2_)_3_N (**1**) (Figure 1) [25].

The widely known practice of adding amines to alkenes (hydroamination), in particular the aza-Michael reaction, which features 100% atom efficiency, meets the requirements of “green” chemistry. It also represents the shortest route to, for example, amino acids and aminoketones, as well as their derivatives, which serve as precursors to the synthesis of various biologically active substances (Figure 2) [41,42,43].

Some aminoalkoxysilanes tolerate the Aza-Michael reaction [44,45]. However, to the best of our knowledge, only one study has reported that 3-aminopropylsilatrane participates in this process [46].

In the present paper, we found that a Michael donor, 3-aminopropylsilatrane **1**, successfully underwent chemical modification (hybridization) via the addition of various alkenes (Michael acceptors), including those that were biologically active. In addition, we evaluated the effect of the synthesized compounds on the growth of microorganisms.

This study focused on:(1)The aza-Michael reaction of 3-aminopropylsilatrane (**1**) with various alkenes;(2)The synthesis and characterization of new 3-aminopropylsilatrane derivatives using IR and NMR spectroscopy and mass and X-ray spectrometry;(3)An in silico investigation of their physical–chemical/pharmacokinetic properties, as well as in vitro evaluations of the inhibitory/stimulating effects on the Gram-positive and Gram-negative pathogenic microorganisms *Listeria*, *Staphylococcus*, and *Yersinia*.

## 2. Results and Discussions

### 2.1. Synthesis

3-Aminopropylsilatrane **1** was synthesized according to the published procedure [24,25,26,46]. The synthesis of new silatrane derivatives **2–7** is shown in Figure 3. The target mono-adducts (**a**) and diadducts (**b**) were obtained via the aza-Michael reaction of **1** with electron-deficient alkenes: acrylonitrile, methylacrylate, ethylacrylate, methyl(meth)acrylate, acrylamide, and N-phenylmaleimide.

To determine the optimum conditions for the process, a number of model experiments were carried out. Among the varied parameters were the ratio of silatrane **1** and acrylonitrile (AN), solvents, temperature, and time. The reaction conditions and the yields of adducts **2a** and **2b** are given in Table 1.

As is seen from Table 1, the best conditions for the synthesis of **2a** and **2b** (99% and 90% yield, respectively) were: methanol as a solvent; 50 °C; 2 h (entry 7 and 8). Notably, diadduct **2b** was formed in the highest yield (99%) in the presence of a small excess of AN for 4 h (entry 9).

The product yields and melting points of silatranes **2**–**7** are presented in Table 2.

Water is known to be efficiently employed as a solvent in the aza-Michael reaction [41]. However, in our case, in the presence of water, adducts were formed in 0% yield. For N-phenylmaleimide, methanol was not suitable because the reaction yielded by-products. Therefore, we used chloroform.

Notably, the reaction of **1** with methyl(meth)acrylate yielded adducts **5a**, **5b**, and **7a** in low yields, while the formation of a diadduct with N-phenylmaleimide was not observed at all, likely due to steric factors.

Thus, the aza-Michael reaction of silatrane **1** with diverse acrylates was studied. It was shown that the interaction proceeded easily and quickly to yield the corresponding mono- or diadducts in high (or good) yields.

At the same time, the reaction of silatrane **1** directly with acrylic acid furnished a mixture of Michael adducts and salt, CH_2_=CH-COO^−^·N^+^H_3_-(CH_2_)_3_-Si(OCH_2_CH_2_)_3_N. Unfortunately, all attempts to separate this mixture failed.

### 2.2. Spectroscopic and Spectrometric Studies

#### 2.2.1. IR Spectroscopy

Silatranes **2**–**7** were investigated using IR spectroscopy (thin film). The IR spectra were recorded at 4000–500 cm^−1^ (Varian 3100). The spectra of silatranes **2–7** show characteristic absorption bands of the silatranyl group at 584–588 cm^−1^ (N→Si), 763–770 cm^−1^, 1090–1103 cm^−1^ (Si-O) and 2925–2939 (ν_s_ CH_2_) [19,21]. The spectra of mono-adducts **2a**–**7a** contain absorption bands (ν_s_ C-NH) and (ν_s_ NH) at 1392–1421 cm^−1^ and 3310–3447 cm^−1^, respectively [46]. In the spectra of silatranes **2a** and **2b**, characteristic absorption bands (C≡N) are observed at 2190–2196 cm^−1^. The spectra of carbonyl-containing mono- and diadducts (silatranes **3**–**7**) display characteristic absorption bands (C=O) at 1671–1736 cm^−1^.

#### 2.2.2. NMR Spectroscopy

The NMR (^1^H, ^13^C, ^29^Si, and ^15^N) spectra of all compounds were recorded at room temperature. The ^1^H NMR spectra show characteristic triplets of the silatranyl moiety, representing NCH_2_ and OCH_2_ groups in the regions of 2.73–2.79 ppm and 3.69–3.76 ppm (*J* = 5.9 Hz), respectively [19,21]. The spectra of all silatranes **2**–**7** exhibit multiplets of the SiCH_2_, CH_2_, and NCH_2_ groups of the propyl moiety at 0.30–0.41 ppm, 1.46–1.61 ppm, and 2.41–2.52 ppm, respectively. In the spectra of all compounds, multiplets of the CH_2_, and CH_2_N groups of ethyl moiety are observed at 2.47–2.52 ppm and 2.84–2.90 ppm, respectively.

In the ^13^C NMR spectra of the synthesized compounds, peaks of the silatranyl moiety carbons (NCH_2_ and OCH_2_) are detected at 50.59–51.21 ppm and 56.30–57.86 ppm, respectively. The spectra of all silatranes show peaks of the propyl moiety carbons (SiCH_2_, CH_2_, and NCH_2_) at 12.80–13.89 ppm, 16.19–24.22 ppm, and 25.15–33.89 ppm, respectively, as well peaks of the ethyl moiety carbons (CH_2_ and CH_2_N) at 52.61–53.01 ppm and 54.01–54.22 ppm, respectively. The spectra of silatranes **2a** and **2b** contain peaks of the nitrile group (C≡N) at 118.42–135.20 ppm. The spectra of carbonyl-containing silatranes display peaks at 173.27–176.53 ppm.

The ^15^N NMR spectra show nitrogen signals at ∼−135 ppm (C≡N), −250 ppm (NH), −340 ppm (N), and −360 ppm (NSilatran).

The ^29^Si NMR spectra of all compounds contain peaks of the silatranyl moiety at ∼−85 ppm. Such δ values indicate an intramolecular transannular Si←N dative bond [19,21].

#### 2.2.3. Mass Spectrometry

High-resolution mass spectra of silatranes **2a** and **2b** were recorded with acetonitrile or methanol as a solvent and perfluorobutyric acid (HFBA) as an ionizing agent. The main ion fragments confirmed the structure of the compounds. The mass spectrum of sample **2a** contained the main peak in the form of the [M + H]^+^ ion. The theoretical *m*/*z* value is 286.158695; in practice, 286.15876 was obtained. The error was 0.2 ppm. The mass spectrum of silatrane **2b** is registered in the form of the ion [M + H]^+^ with *m*/*z* 339.18498 (theoretical *m*/*z* value is 339.185244); the error is 0.8 ppm.

#### 2.2.4. X-ray Spectrometry

X-ray crystallography. Single crystals of **2a** and **2b** were grown from CHCl_3_ solution. The molecular structures are depicted in Figure 2. The selected bonds lengths and angles are given in Table 3.

### 2.3. Study of Biological Activity

Before examining the biological, physiological, and pharmacological activity of compounds in vitro, it is always useful to evaluate their physicochemical properties in silico.

#### 2.3.1. In Silico Evaluation of Physicochemical and Pharmacokinetic Properties as Well as Pharmacological Activity

Bioavailability of the obtained compounds **2**–**7** was evaluated using the Lipinski rule [47]. The ADME (absorption, distribution, metabolism, and excretion) properties were studied using SwissADME online software (http://www.swissadme.ch accessed on 20 April 2023). The calculations show that the properties of the synthesized silatranes agree with the Lipinski rule: M.W. < 500, lipophilicity (octanol–water partition coefficient, Log *p* < 5), a number of hydrogen bond acceptors, HBA < 10 and a number of hydrogen bond donors, HBD < 5) and can be considered as druglike products [47].

Compounds **2–7** are highly soluble in water (3–4 mg/mL or 7–8 mol/L) and possess gastrointestinal absorption (score—High) as well as bioavailability (0.55) properties. Probable pharmacological activity profiles of silatranes **2**–**7** were studied in silico using PASS 2.0 software [48,49]. The calculated screening showed that the obtained silatranes exhibit pronounced antineoplastic activity (probability 98–99%). Compounds **2**–**7** can inhibit the activity of specific enzymes, such as saccharopepsin (75–77%), gluconate 2-dehydrogenase (70–75%), and CDP-glycerol glycerophosphotransferase (70%). On the other hand, these silatranes can stimulate macrophage colonies (probability, 80%).

Thus, the values calculated using SwissADME and PASS confirm that the synthesized silatranes are good candidates for further research.

#### 2.3.2. Study of the Inhibitory/Stimulating Effect on Microorganisms In Vitro

Silatranes are known to affect the growth and development of animals, insects, plants and microorganisms. In particular, they can stimulate or inhibit the growth of some fungi and bacteria [21,25]. The available data indicate that silatranes induce changes in the metabolic systems of microorganisms, which increase or decrease the resistance of bacteria.

The membranotropism of silatranes, determined by their high lipid solubility (see, for example, Section 2.3.1), water solubility, bioavailability, and the ability to penetrate through cell membranes due to the interaction with polar groups of proteins and lipids are valuable properties of these compounds in the mechanism of biological activity. In addition, the substituent at the silicon atom plays an important significant role [23,25,31,32,33,34]. Finally, the biological activity of silatranes may depend on the concentration of the active substance, as well as on the type of microorganism.

Microbiological studies were carried out in the Irkutsk Antiplague Research Institute of Siberia and Far East (IARI). Test cultures were pathogenic Gram-positive and Gram-negative bacterial strains (from the IARI collection). Gram-positive stains included causative agent of listeriosis, *Listeria monocytogenes* 766 and infectious agent, *Staphylococcus aureus* ATCC 6538-P (FDA 209-P). Gram-negative strains were causative agent of yersiniosis, *Yersinia enterocolitica* 03 628/1; causative agent of scarletlike fever, *Yersinia pseudotuberculosis* 223; and causative agent of plague, *Yersinia pestis* EV NIIEG (Collection of Research Institute of Epidemiology and Hygiene). Test strains were cultivated on Hottinger’s agar (HA).

To begin with, we evaluated the effect of selected silatranes **2a**, **2b**, **4a**, **5a**, and **7a** at a concentration of 400–200 μg/mL on the growth of microorganisms using a method of broth serial dilution and a method of diffusion into agar [50]. It was shown that the studied compounds insignificantly inhibited the growth of all microorganisms (Inhibition Index—II by an average of 33.7 ± 9.5%). At a lower concentration (100 µg/mL), silatranes **2a**, **2b**, **4a**, **5a**, and **7a** even more weakly inhibited the growth of test strains *L. monocytogenes*, *Y. pestis*, *Y. enterocolitica*, and Y. pseudotuberculosis (II by an average of 16.2 ± 8.5 %). In the experiments using the diffusion method into HA, no zones of inhibition were observed.

When the concentration of samples was consistently decreased further (up to 6 and up to 3 µg/mL), we serendipitously found that the tested silatranes exerted non-inhibitory, but stimulating, activity (Stimulation Index, SI, up to 76%) (Table 4 and Figure 3).

As shown in Table 4 and Figure 3, mono-**2a** and diadduct **2b** bearing nitrile groups (C≡N) showed the best stimulatory activity with respect to Gram-positive microorganisms (*S. aureus*). Notably, compound **2a** turned out to be much more efficient. Amino-acid-like silatranes **4a**, **5a**, and **7a** were active against Gram-negative *Yersinia* organisms. At the same time, the more branched compounds, **5a** and **7a**, were less effective.

Sample **4a**, which exhibits stimulating activity against all studied microorganisms, was selected for further experiments. Figure 4 shows the growth properties of *S. aureus* and *Y. pseudotuberculosis* cultivated on the HA nutrient medium without (control) and with (experiment) the addition of **4a**.

The data in Figure 4 also confirm that the cultivation of microorganisms on HA with the addition of silatrane **4a** was ~2 times more effective than with the control.

Thus, the synthesized silatranes can be used as a promising stimulating agents and bacterial biomass growth promoters for the accelerated analysis and diagnosis of diseases caused by *Listeria*, *Staphylococcus*, *Yersinia*, and other pathogens.

## 3. Materials and Methods

### 3.1. Chemistry

The organic solvents were dried and purified using standard procedures. Acrylonitrile, methylacrylate, ethylacrylate, methyl(meth)acrylate, acrylamide, and N-phenylmaleimide were purchased from Sigma Aldrich.

Target compounds **2**–**7** were isolated in pure form via recrystallization from a mixture of chloroform/hexane (1:3).

IR spectra were registered on a Varian 3100 FTIR spectrometer in the 4000–400 cm^−1^ range with the sample as a thin film or tablet (KBr).

^1^H, ^13^C, and ^15^N NMR spectra were performed in CDCl_3_ at room temperature on Bruker DPX-400 and AV-400 spectrometers (400.13, 100.61, and 40.56 MHz, respectively).

Chemical shifts were referred to TMS (^1^H and ^13^C) and nitromethane (^15^N).

Mass spectra were recorded on HR-TOF-ESI-MS Agilent 6210 equipment with the registration mode of positive ions with acetonitrile as a solvent (in case of poor solubility via ultrasound) and 0.1% perfluorobutyric acid as an ionizing agent.

To perform X-ray diffraction analysis, single crystals of compounds **2a** and **2b** were grown via the slow evaporation of chloroform solutions at room temperature. The X-ray diffraction data were collected with a Bruker D8 VENTURE diffractometer (PHOTON III CMOS detector, Mo IµS3.0 X-ray source, Montel mirror-focused MoKα radiation λ = 0.71073 Å, N_2_-flow cryostat) via 0.5° ω- and φ-scan techniques. Data were corrected for absorption effects using the multi-scan method (SADABS) [51]. The structure was solved and refined using the Bruker SHELXTL Software Package (Sheldrick, 2008) [52].

The major non-H atoms were located from the electron density map and refined in anisotropic approximation. The positions of minor conformation (ca. 13% occupancy) of the disordered silotrane fragment in the **2b** crystal structure and all H atoms of ordered parts of the structures were located from the difference electron density map. The atoms of the minor conformation were refined isotropically with equal atomic displacements (EADP) and restraints on the similarity of equivalent bond distances (SADI). The H atom located on N2 in **2a** was refined without geometrical restraints; the other H atoms were refined in the riding model. Isotropic displacements of the H atoms were assigned as 1.2Ueq of the pivot atoms.

Crystal data, data collection, and structure refinement details are summarized in Table 5. Atomic coordinates, bond lengths, bond angles, and atomic displacement parameters for the crystal structures of **2a** and **2b** have been deposited with the Cambridge Crystallographic Data Centre (CCDC) with deposition numbers CCDC 2263862 and 2263863. These data can be obtained free of charge from the CCDC via URL: https://www.ccdc.cam.ac.uk/structures/ (accessed on 18 May 2023).

Elemental analysis was performed on a Thermo Scientific Flash 2000 Elemental Analyzer (Thermo Fisher Scientific Inc., Milan, Italy). Melting points were determined on a Kofler Hot-Stage Microscope PolyTherm A apparatus (Wagner & Munz GmbH, München, Germany).

General Procedure for the Synthesis of Silatranes **2**–**7**.

A mixture of silatrane **1** (1 mmol) and corresponding acrylate (1 or 2 mmol) in 10 mL of methanol was stirred at 50 °C for 2 h in an inert atmosphere (N_2_) or in air. The solvent was removed under reduced pressure. The residue was washed many times with ether, dried, and the products **2–7** were obtained. If necessary, the resulting powder or oil was recrystallized from a mixture of chloroform/hexane (1:3).

#### 3.1.1. Silatrane **2a**

The product was recrystallized from a chloroform/hexane (1:3) mixture at 4 °C, 24 h. Needle crystals.

Anal. calc. for C_12_H_23_N_3_O_3_Si (%): C, 50.49; H, 8.12; N, 14.72. Found (%): C, 50.68; H, 8.09;

N, 14.89.

IR (ν, cm^−1^): 585 (N→Si), 763, 1101 (Si-O), 1416 (ν_s_ C-NH), 2190 (C≡N), 2925 (ν_s_ CH_2_),

3310 (ν_s_ NH).

^1^H-NMR (CDCl_3_, 400 MHz), δ (ppm): 0.35–0.39 (m, 2H, SiCH_2_); 1.52–1.56 (m, 2H, SiCH_2_CH_2_); 2.44 (t, J = 5.5 Hz, 2H, CH_2_CN); 2.58–2.63 (m, 2H, SiCH_2_CH_2_CH_2_); 2.76 (t, J = 5.3 Hz, 6H, NCH_2_); 2.87–2.91 (m, 2H, CH_2_CH_2_CN); 3.61 (br.s, 1H, NH); 3.71 (t, J = 5.3 Hz, 6H, OCH_2_).

^13^C-NMR (CDCl_3_, 100 MHz), δ (ppm): 13.36 (SiCH_2_); 18.77 (CH_2_CN); 25.15 (SiCH_2_CH_2_); 44.97 (CH_2_CH_2_CN); 51.20 (NCH_2_); 52.33 (SiCH_2_CH_2_CH_2_); 57.83 (OCH_2_); 118.99 (CN).

^15^N-NMR (CDCl_3_, 40 MHz), δ (ppm): −359.6 (N_Sil_); −343.4 (NH); −135.2 (CN).

HR-MS: *m/z* = 286.15876 [(M + H), 100].

#### 3.1.2. Silatrane **2b**

The product was recrystallized from a chloroform/hexane (1:3) mixture at 4 °C, 24 h.

Large transparent crystals.

Anal. calc. for C_15_H_26_N_4_O_3_Si (%): C, 53.23; H, 7.74; N, 16.55. Found (%): C, 53.54; H, 7.58; N, 16.69.

IR (ν, cm^−1^): 584 (N→Si), 764, 1103 (Si-O), 2196 (C≡N), 2930 (ν_s_ CH_2_).

^1^H-NMR (CDCl_3_, 400 MHz), δ (ppm): 0.35–0.39 (m, 2H, SiCH_2_); 1.52–1.56 (m, 2H, SiCH_2_CH_2_); 2.40–2.46 (m, 6H, CH_2_CN, SiCH_2_CH_2_CH_2_); 2.78 (t, J = 5.3 Hz, 6H, NCH_2_); 2.80–2.86 (m, 4H, CH_2_CH_2_CN); 3.74 (t, J = 5.3 Hz, 6H, OCH_2_).

^13^C-NMR (CDCl_3_, 100 MHz), δ (ppm): 13.22 (SiCH_2_); 18.05 (CH_2_CN); 24.63 (SiCH_2_CH_2_); 47.92 (CH_2_CH_2_CN); 51.25 (NCH_2_); 57.07 (SiCH_2_CH_2_CH_2_); 57.80 (OCH_2_); 118.92 (CN).

^15^N-NMR (CDCl_3_, 40 MHz), δ (ppm): −359.6 (N_Sil_); −341.4 (NH); −135.3 (CN).

HR-MS: *m/z* = 339.18498 [(M + H), 100].

#### 3.1.3. Silatrane **3a**

Anal. calc. for C_13_H_26_N_2_O_5_Si (%): C, 49.03; H, 8.23; N, 8.79. Found (%): C, 50.15; H, 8.12;

N, 8.96.

IR (ν, cm^−1^): 586 (N→Si), 764, 1100 (Si-O), 1419 (ν_s_ C-NH), 1736 (C=O), 2933 (ν_s_ CH_2_), 3447 (ν_s_ NH).

^1^H-NMR (CDCl_3_, 400 MHz), δ (ppm): 0.26–0.28 (m, 2H, SiCH_2_); 1.46–1.50 (m, 2H, SiCH_2_CH_2_); 2.48–2.52 (m, 2H, SiCH_2_CH_2_CH_2_); 2.52 (t, J = 6.6 Hz, 2H, CH_2_CH_2_CO); 2.77 (t, J = 5.3 Hz, 6H, NCH_2_); 2.84 (t, J = 6.6 Hz, 2H, CH_2_CH_2_CO); 3.22 (br.s, 1H, NH); 3.60 (s, 3H, OMe); 3.75 (t, J = 5.3 Hz, 6H, OCH_2_).

^13^C-NMR (CDCl_3_, 100 MHz), δ (ppm): 13.61 (SiCH_2_); 25.35 (SiCH_2_CH_2_); 34.92 (CH_2_CH_2_CO); 46.18 (CH_2_CH_2_CO); 51.28 (NCH_2_); 51.36 (OMe); 52.74 (SiCH_2_CH_2_CH_2_); 57.79 (OCH_2_); 173.32 (CO).

^15^N-NMR (CDCl_3_, 40 MHz), δ (ppm): −359.3 (N_Sil_); −342.2 (NH).

#### 3.1.4. Silatrane **3b**

Anal. calc. for C_17_H_32_N_2_O_7_Si (%): C, 50.47; H, 7.97; N, 6.92. Found (%): C, 50.54; H, 7.74;

N, 6.99.

IR (ν, cm^−1^): 587 (N→Si), 764, 1090 (Si-O), 1721 (C=O), 2936 (ν_s_ CH_2_).

^1^H-NMR (CDCl_3_, 400 MHz), δ (ppm): 0.25–0.29 (m, 2H, SiCH_2_); 1.44–1.48 (m, 2H, SiCH_2_CH_2_); 2.35–2.37 (m, 2H, SiCH_2_CH_2_CH_2_); 2.41 (t, J = 6.6 Hz, 4H, CH_2_CH_2_CO); 2.73 (t, J = 6.6 Hz, 4H, CH_2_CH_2_CO); 2.76 (t, J = 5.3 Hz, 6H, NCH_2_); 3.62 (s, 6H, OMe); 3.74 (t, J = 5.3 Hz, 6H, OCH_2_).

^13^C-NMR (CDCl_3_, 100 MHz), δ (ppm): 13.55 (SiCH_2_); 24.80 (SiCH_2_CH_2_); 32.70 (CH_2_CH_2_CO); 49.21 (CH_2_CH_2_CO); 51.22 (NCH_2_); 51.38 (OMe); 57.28 (SiCH_2_CH_2_CH_2_); 57.81 (OCH_2_); 173.27 (CO).

^15^N-NMR (CDCl_3_, 40 MHz), δ (ppm): −359.4 (N_Sil_); −339.9 (NCH_2_).

#### 3.1.5. Silatrane **4a**

Anal. calc. for C_14_H_28_N_2_O_5_Si (%): C, 50.57; H, 8.49; N, 8.42. Found (%): C, 50.75; H, 8.26;

N, 8.69.

IR (ν, cm^−1^): 588 (N→Si), 768, 1098 (Si-O), 1421 (ν_s_ C-NH), 1720 (C=O), 2925 (ν_s_ CH_2_), 3430 (ν_s_ NH).

^1^H-NMR (CDCl_3_, 400 MHz), δ (ppm): 0.26–0.28 (m, 2H, SiCH_2_); 1.25 (t, J = 7.1 Hz, 3H, OCH_2_CH_3_); 1.45–1.49 (m, 2H, SiCH_2_CH_2_); 2.48 (t, J = 6.6 Hz, 2H, CH_2_CH_2_CO); 2.50–2.54 (m, 2H, SiCH_2_CH_2_CH_2_); 2.78 (t, J = 5.3 Hz, 6H, NCH_2_); 2.89 (t, J = 6.6 Hz, 2H, CH_2_CH_2_CO); 3.10 (br.s, 1H, NH); 3.75 (t, J = 5.3 Hz, 6H, OCH_2_); 4.13 (q, J = 7.1 Hz, 2H, OCH_2_CH_3_).

^13^C-NMR (CDCl_3_, 100 MHz), δ (ppm): 13.20 (SiCH_2_); 13.82 (OCH_2_CH_3_); 25.32 (SiCH_2_CH_2_); 34.52 (CH_2_CH_2_CO); 47.12 (CH_2_CH_2_CO); 51.22 (NCH_2_); 52.63 (SiCH_2_CH_2_CH_2_); 57.75 (OCH_2_); 59.72 (OCH_2_CH_3_); 172.13 (CO).

^15^N-NMR (CDCl_3_, 40 MHz), δ (ppm): −359.7 (N_Sil_); −343.2 (NH).

#### 3.1.6. Silatrane **4b**

Anal. calc. for C_19_H_36_N_2_O_7_Si (%): C, 52.75; H, 8.38; N, 6.48. Found (%): C, 52.94; H, 8.21;

N, 6.59.

IR (ν, cm^−1^): 587 (N→Si), 770, 1100 (Si-O), 1717 (C=O) 2930 (ν_s_ CH_2_).

^1^H-NMR (CDCl_3_, 400 MHz), δ (ppm): 0.26–0.28 (m, 2H, SiCH_2_); 1.25 (t, J = 7.1 Hz, 6H, OCH_2_CH_3_); 1.45–1.49 (m, 2H, SiCH_2_CH_2_); 2.37–2.41 (m, 2H, SiCH_2_CH_2_CH_2_); 2.44 (t, J = 6.6 Hz, 4H, CH_2_CH_2_CO); 2.76 (t, J = 5.3 Hz, 6H, NCH_2_); 2.75 (t, J = 6.6 Hz, 4H, CH_2_CH_2_CO); 3.77 (t, J = 5.3 Hz, 6H, OCH_2_); 4.11 (q, J = 7.1 Hz, 4H, OCH_2_CH_3_).

^13^C-NMR (CDCl_3_, 100 MHz), δ (ppm): 13.25 (SiCH_2_); 13.85 (OCH_2_CH_3_); 24.90 (SiCH_2_CH_2_); 32.48 (CH_2_CH_2_CO); 49.27 (CH_2_CH_2_CO); 51.27 (NCH_2_); 57.71 (SiCH_2_CH_2_CH_2_); 57.73 (OCH_2_); 59.83 (OCH_2_CH_3_); 172.22 (CO).

^15^N-NMR (CDCl_3_, 40 MHz), δ (ppm): −359.9 (N_Sil_); −341.5 (NCH_2_).

#### 3.1.7. Silatrane **5a**

Anal. calc. for C_14_H_28_N_2_O_5_Si (%): C, 50.57; H, 8.49; N, 8.42. Found (%): C, 50.68; H, 8.32;

N, 8.63.

IR (ν, cm^−1^): 583 (N→Si), 756, 1097 (Si-O), 1456 (ν_s_ C-NH), 1726 (C=O), 2924 (ν_s_ CH_2_), 3435 (ν_s_ NH).

^1^H-NMR (CDCl_3_, 400 MHz), δ (ppm): 0.28–0.31 (m, 2H, SiCH_2_); 1.07 (d, J = 6.7 Hz, 3H, CHCH_3_); 1.47–1.52 (m, 2H, SiCH_2_CH_2_); 2.50–2.54 (m, 3H, CH_2_CHCH_3_, SiCH_2_CH_2_CH_2_); 2.55–2.59 (m, 1H, CHCH_3_); 2.73 (t, J = 5.3 Hz, 6H, NCH_2_); 2.75–2.79 (m, 1H, CH_2_CHCH_3_); 3.10 (br.s, 1H, NH); 3.60 (s, 3H, OCH_3_); 3.69 (t, J = 5.3 Hz, 6H, OCH_2_).

^13^C-NMR (CDCl_3_, 100 MHz), δ (ppm): 13.43 (SiCH_2_); 15.32 (CHCH_3_); 25.28 (SiCH_2_CH_2_); 40.13 (CHCH_3_); 52.66 (CH_2_CHCH_3_); 51.21 (NCH_2_); 51.56 (OCH_3_); 53.21 (SiCH_2_CH_2_CH_2_); 57.86 (OCH_2_); 176.53 (CO).

^15^N-NMR (CDCl_3_, 40 MHz), δ (ppm): −359.7 (N_Sil_); −343.6 (NH).

#### 3.1.8. Silatrane **5b**

Anal. calc. for C_19_H_36_N_2_O_7_Si (%): C, 52.75; H, 8.38; N, 6.48. Found (%): C, 52.88; H, 8.27;

N, 6.54.

IR (ν, cm^−1^): 587 (N→Si), 770, 1100 (Si-O), 1715 (C=O) 2935 (ν_s_ CH_2_).

^1^H-NMR (CDCl_3_, 400 MHz), δ (ppm): 0.28–0.31 (m, 2H, SiCH_2_); 1.05 (d, J = 6.7 Hz, 6H, CHCH_3_); 1.47–1.52 (m, 2H, SiCH_2_CH_2_); 2.35–2.39 (m, 2H, SiCH_2_CH_2_CH_2_); 2.46–2.49 (m, 2H, CH_2_CHCH_3_); 2.50–2.53 (m, 2H, CHCH_3_); 2.71 (t, J = 5.3 Hz, 6H, NCH_2_); 2.63–2.66 (m, 2H, CH_2_CHCH_3_); 3.58 (s, 6H, OCH_3_); 3.71 (t, J = 5.3 Hz, 6H, OCH_2_).

^13^C-NMR (CDCl_3_, 100 MHz), δ (ppm): 13.49 (SiCH_2_); 15.30 (CHCH_3_); 24.83 (SiCH_2_CH_2_); 39.15 (CHCH_3_); 54.31 (CH_2_CHCH_3_); 51.23 (NCH_2_); 51.52 (OCH_3_); 58.06 (SiCH_2_CH_2_CH_2_); 57.89 (OCH_2_); 176.68 (CO).

^15^N-NMR (CDCl_3_, 40 MHz), δ (ppm): −359.7 (N_Sil_); −340.2 (NCH_2_).

#### 3.1.9. Silatrane **6a**

Anal. calc. for C_12_H_25_N_3_O_4_Si (%): C, 47.49; H, 8.30; N, 13.85. Found (%): C, 47.61; H, 8.19; N, 13.97.

IR (ν, cm^−1^): 586 (N→Si), 770, 1100 (Si-O), 1419 (ν_s_ C-NH), 1671 (C=O), 2930 (ν_s_ CH_2_), 3194 (NH_2_), 3317 (ν_s_ NH).

^1^H-NMR (CDCl_3_, 400 MHz), δ (ppm): 0.28–0.31 (m, 2H, SiCH_2_); 1.46–1.50 (m, 2H, SiCH_2_CH_2_); 2.52–2.56 (m, 2H, SiCH_2_CH_2_CH_2_); 2.63 (t, J = 6.3 Hz, 2H, CH_2_CH_2_CO); 2.77 (t, J = 5.3 Hz, 6H, NCH_2_); 2.87 (t, J = 6.3 Hz, 2H, CH_2_CH_2_CO); 3.15 (br.s, 1H, NH); 3.75 (t, J = 5.3 Hz, 6H, OCH_2_); 5.52 (br.s., 1H, NH_2_); 7.84 (br.s., 1H, NH_2_).

^13^C-NMR (CDCl_3_, 100 MHz), δ (ppm): 13.31 (SiCH_2_); 25.11 (SiCH_2_CH_2_); 35.27 (CH_2_CH_2_CO); 45.26 (CH_2_CH_2_CO); 51.28 (NCH_2_); 51.88 (SiCH_2_CH_2_CH_2_); 57.79 (OCH_2_); 175.62 (CO).

^15^N-NMR (CDCl_3_, 40 MHz), δ (ppm): −359.9 (N_Sil_); −342.7 (NH); −267.3 (NH_2_).

#### 3.1.10. Silatrane **6b**

Anal. calc. for C_19_H_36_N_2_O_7_Si (%): C, 48.11; H, 8.07; N, 14.96. Found (%): C, 48.31; H, 7.92; N, 15.12.

IR (ν, cm^−1^): 587 (N→Si), 770, 1100 (Si-O), 1681 (C=O), 2931 (ν_s_ CH_2_).

^1^H-NMR (CDCl_3_, 400 MHz), δ (ppm): 0.28–0.31 (m, 2H, SiCH_2_); 1.44–1.48 (m, 2H, SiCH_2_CH_2_); 2.40–2.44 (m, 2H, SiCH_2_CH_2_CH_2_); 2.46 (t, J = 6.3 Hz, 4H, CH_2_CH_2_CO); 2.73 (t, J = 6.3 Hz, 4H, CH_2_CH_2_CO); 2.76 (t, J = 5.3 Hz, 6H, NCH_2_); 3.74 (t, J = 5.3 Hz, 6H, OCH_2_); 6.10 (br.s., 2H, NH_2_); 7.63 (br.s., 2H, NH_2_).

^13^C-NMR (CDCl_3_, 100 MHz), δ (ppm): 13.34 (SiCH_2_); 24.83 (SiCH_2_CH_2_); 33.14 (CH_2_CH_2_CO); 49.37 (CH_2_CH_2_CO); 51.22 (NCH_2_); 57.12 (SiCH_2_CH_2_CH_2_); 57.81 (OCH_2_); 175.83 (CO).

^15^N-NMR (CDCl_3_, 40 MHz), δ (ppm): −359.2 (N_Sil_); −339.4 (NCH_2_); −267.3 (NH_2_).

#### 3.1.11. Silatrane **7a**

Anal. calc. for C_18_H_25_N_3_O_5_Si (%): C, 55.22; H, 6.44; N, 10.73. Found (%): C, 55.45; H, 6.28; N, 10.91.

IR (ν, cm^−1^): 582 (N→Si), 759, 1097 (Si-O), 1391 (ν_s_ C-NH), 1712 (C=O), 2932 (ν_s_ CH_2_), 3467 (ν_s_ NH).

^1^H-NMR (CDCl_3_, 400 MHz), δ (ppm): 0.38–0.41 (m, 2H, SiCH_2_); 1.52–1.56 (m, 2H, SiCH_2_CH_2_); 2.23 (br.s, 1H, NH); 2.62–2.66 (m, 3H, SiCH_2_CH_2_CH_2_, CH_2_CH); 2.73 (t, J = 5.3 Hz, 6H, NCH_2_); 2.99 (dd, J = 17.8 Hz, J = 8.3 Hz, 1H, CH_2_CH); 3.69 (t, J = 5.3 Hz, 6H, OCH_2_); 3.89 (dd, J = 8.3 Hz, J = 5.6 Hz, 1H, CH_2_CH); 7.20–7.24 (m, 2H. Ph); 7.42–7.48 (m, 3H, Ph).

^13^C-NMR (CDCl_3_, 100 MHz), δ (ppm): 13.47 (SiCH_2_); 24.95 (SiCH_2_CH_2_); 36.38 (CH_2_CH); 50.54 (SiCH_2_CH_2_CH_2_); 50.82 (NCH_2_); 56.02 (CH_2_CH); 57.50 (OCH_2_); 126.32; 128.43; 128.97; 131.68 (Ph); 174.60 (CO); 177.04 (CO).

^15^N-NMR (CDCl_3_, 40 MHz), δ (ppm): −359.5 (N_Sil_); −337.6 (NH); −192.4 (NPh).

### 3.2. Biology

#### Bacterial Strains; Microbiology Testing

Test cultures included pathogenic Gram-positive and Gram-negative bacterial strains (Collection of Irkutsk Antiplague Research Institute of Siberia and Far East, IARI): *Listeria monocytogenes* 766, *Staphylococcus aureus* ATCC 6538-P (FDA 209-P), *Yersinia enterocolitica* 03 628/1, *Yersinia pseudotuberculosis* 223, and *Yersinia pestis* EV. Test strains *Staphylococcus* and *Yersinia* were cultivated on Hottinger’s agar (HA) at (28 ± 1) °C for 48 h. *Listeria* were grown on an HA or Hottinger’s broth with 1% glucose (Himedia, India, pH 7.2) and incubated for 24 h at (37 ± 1) °C.

To determine the sensitivity of microorganisms to the selected silatranes **2a**, **2b**, **4a**, **5a**, and **7a**, the diffusion method and the method of serial dilutions in broth were used [50]. Growth medium (100 μL) containing 400–3.125 μg mL^−1^ of test compounds (in H_2_O) was added to a 96-well microtiter plate. Culture suspensions were diluted to the 0.5 McFarland standard (1.5 × 10^8^ CFU mL^−1^). The microbial suspension was diluted to 1:10 (up to 1 × 10^5^ CFU mL^−1^), and 50 μL was added to each well. The plates were incubated for 24 h and the optical density (OD) was measured at λ = 490 nm (xMark Spectrophotometer 1681150, Bio-Rad, Foster City, CA, USA). As a negative control, the growth medium was used without the addition of test compounds. The standard aminoglycoside antibiotic Gentamicin (positive control) was used for comparison.

The inhibitory/stimulating effect of silatranes was determined by measuring the optical density (OD) after 24–48 h of incubation. The results were obtained on an automatic reader for the xMark microplate according to a unified method and expressed as an Inhibition/Stimulation index (II/SI) value of microorganism growth, which was calculated as the ratio of the OD values in the experimental (ODexp) and control (ODcontr) samples and expressed as a percentage by the following formula: II (SI) = ODexp − ODcontr/ODcontr × 100%.

The specific activity of nutrient media (germination coefficient, medium sensitivity, growth rate, and cultural–morphological properties of microorganisms) was assessed using a complex of microbiological methods. From a suspension of 1 × 10^9^ CFU/mL, 10-fold dilutions (10^−3^–10^−8^) were prepared; 0.1 mL of culture suspension from dilutions 10^−6^ and 10^−7^ were inoculated three times per Petri dish with a nutrient medium. The suspension was evenly distributed over the surface of the HA plate. The nutrient medium without the addition of silatranes was used as a control. Inoculation plates were examined after 16, 24, and 48 h of incubation at a temperature corresponding to the microorganism.

On HA after incubation, colonies of microorganism test strains were counted, and the germination coefficient, Cger, was determined according to the formula: Cger = Ns/Ncontr, where Ns is the number of colonies on HA with the addition of silatranes, and Ncontr is the number of colonies on the control nutrient medium. The acceptance criterion for the experimental culture medium was if the germination coefficient was 0.5–2, it was considered suitable for use when compared with the control culture medium.

All results were processed using statistically standard methods in the Microsoft Excel 14.0 software package. Data were expressed as the arithmetic mean (M) and standard deviation (s). Differences were taken as significant at a significance level of *p* < 0.05.

## 4. Conclusions

In conclusion, it is found that the aza-Michael reaction of 3-aminopropylsilatrane with various acrylates, depending on the reagent ratio, yields the corresponding mono- and diadducts. This reaction proceeds under mild conditions to afford the functionalized silatranes in high yields. The structures of the target compounds were characterized using FT-IR, NMR spectroscopy, high-resolution mass spectrometry, X-ray diffraction, and elemental analysis.

In silico computational screening of the synthesized silatranes using SwissADME and PASS 2.0 software revealed that these compounds are synthetically accessible, bioavailable, are druglike, and exhibit antineoplastic (probability 98–99%) and macrophage colony stimulating (probability 80%) activity.

The in vitro study demonstrated that silatranes at low concentrations (3–6 µg/mL) exhibit a pronounced stimulating effect (up to 76%) on the growth of pathogenic microorganisms (*Listeria*, *Staphylococcus*, and *Yersinia*) and can be used as biomass growth stimulants for accelerated analysis and express diagnostics of diseases caused by these bacteria.

In general, silatranes had a selective effect on the studied bacteria. These data inspire the further identification of specific drugs or their analogues that can be used as effective inhibitors or stimulators of the vital activity of microorganisms, and their optimal concentrations.

## Data Availability

The data presented in this study are available from the authors upon reasonable request.

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
