# Peer review of "New 3-Aminopropylsilatrane Derivatives: Synthesis, Structure, Properties, and Biological Activity"

_ijms, 2023, doi:10.3390/ijms24129965_

Round 1
Reviewer 1 Report
This paper can not accepted in its current form.
Important biological data doesn't have error bars.
Synthetic schemes, missing the reagents and conditions.
NMR spectra are not shown for new molecules.
Paper is overall poorly written.
Author Response
Important biological data doesn't have error bars.
Answer:
We did not use error bars on the figures due to the small scale, which we had to apply to the very large Figure 3. However, we expressed other important error bars as percentages, for example, by an average of 33.7 ± 9.5%, and 16.2 ± 8.5% (marked green in the text). In addition, Table 4 shows the corresponding minimum and maximum values of Stimulation Index (SI).
Synthetic schemes, missing the reagents and conditions.
Answer:
Synthetic schemes and their captions have been revised (marked green in the text).
NMR spectra are not shown for new molecules.
Answer:
The complete description of all 1H, 13C, and 15N spectra is given in the text of the paper. In addition, IR and mass spectra are discussed in detail. X-ray diffraction data are included for 2 new compounds. Making a special Supplementary Information (SI) file, as a rule, takes a lot of time and effort. Besides, as we were confirmed by the editors of Int. J. Mol. Sci., according to the journal guidelines, the inclusion of the SI file is optional.
However, if the reviewer will insist, we can include 2-3 NMR spectra directly in the text of the paper.
Reviewer 2 Report
The compound 3-aminopropylsilatrane can react with various acrylates and Michael acceptors, resulting in the formation of mono- or diadducts with different functional groups. These functionalized silatranes have been extensively characterized and show promising bioavailability as drug-like compounds with antineoplastic and macrophage colony stimulating properties, as well as inhibitory and stimulating effects on pathogenic bacteria growth depending on their concentration.
The paper is well written; however, it is recommended to consider including the IR and NMR spectra in the Supplementary Information (SI) to provide a more complete and detailed characterization of the synthesized compounds. This addition would further strengthen the scientific rigor and enhance the overall quality of the publication.
Author Response
The paper is well written; however, it is recommended to consider including the IR and NMR spectra in the Supplementary Information (SI) to provide a more complete and detailed characterization of the synthesized compounds. This addition would further strengthen the scientific rigor and enhance the overall quality of the publication.
Answer:
The complete description of all 1H, 13C, and 15N spectra is given in the text of the paper. In addition, IR and mass spectra are discussed in detail. X-ray diffraction data are included for 2 new compounds. Making a special Supplementary Information (SI) file, as a rule, takes a lot of time and effort. Besides, as we were confirmed by the editors of Int. J. Mol. Sci., according to the journal guidelines, the inclusion of the SI file is optional.
However, if the reviewer will insist, we can include 2-3 NMR spectra directly in the text of the paper.

Round 2
Reviewer 1 Report
The paper can be accepted.